# Immune-Related Adverse Events, Biomarkers of Systemic Inflammation, and Survival Outcomes in Patients Receiving Pembrolizumab for Non-Small-Cell Lung Cancer

**DOI:** 10.3390/cancers15235502

**Published:** 2023-11-21

**Authors:** George Raynes, Mark Stares, Samantha Low, Dhania Haron, Hussain Sarwar, Dhruv Abhi, Colin Barrie, Barry Laird, Iain Phillips, Melanie MacKean

**Affiliations:** 1Edinburgh Cancer Centre, NHS Lothian, Western General Hospital, Crewe Road South, Edinburgh EH4 2XU, UK; 2Cancer Research UK Scotland Centre, Institute of Genetics and Cancer, Western General Hospital, University of Edinburgh, Crewe Road South, Edinburgh EH4 2XR, UK

**Keywords:** biomarkers of systemic inflammation, Scottish Inflammatory Prognostic Score (SIPS), non-small-cell lung cancer, immune-related adverse events, pembrolizumab

## Abstract

**Simple Summary:**

Immune checkpoint inhibitors offer the chance for the durable disease control of advanced/metastatic non-small-cell lung cancer (NSCLC). However, they come with the risk of immune-related adverse events (irAEs) which may be severe or life-threatening, or lead to long-term toxicity. We confirm that the occurrence of irAEs is associated with improved survival outcomes in patients with NSCLC treated with pembrolizumab. These survival benefits are only seen in patients with particular irAEs, or mild irAEs. Significantly, we account for the time-dependent association between time on treatment and the development of irAEs. Low levels of systemic inflammation, as defined by simple biomarkers of systemic inflammation, have previously been shown to predict the occurrence of irAEs. However, we find that their predictive power is confounded by their independent prognostic value. We suggest that these findings are taken into account by other studies investigating potential biomarkers of irAE risk.

**Abstract:**

Background: Pembrolizumab monotherapy for non-small-cell lung cancer (NSCLC) expressing PD-L1 ≥ 50% doubles five-year survival rates compared to chemotherapy. However, immune-related adverse events (irAEs) can cause severe, long-term toxicity necessitating high-dose steroids and/or treatment cessation. Interestingly, patients experiencing irAEs demonstrate better survival outcomes. Biomarkers of systemic inflammation, including the Scottish Inflammatory Prognostic Score (SIPS), also predict survival in this patient group. This study examines the relationship between inflammatory status, irAEs, and survival outcomes in NSCLC. Methods: A retrospective analysis was conducted on patients with NSCLC expressing PD-L1 ≥ 50% receiving first-line pembrolizumab monotherapy at a large cancer centre in Scotland. Regression analyses were conducted to examine the relationship between SIPS, irAEs, and survival. Results: 83/262 eligible patients (32%) experienced an irAE. Dermatological, endocrine, gastrointestinal, and hepatic, but not pulmonary, irAEs were associated with prolonged PFS and OS (*p* <= 0.011). Mild irAEs were associated with better PFS and OS in all patients, including on time-dependent analyses (HR0.61 [95% CI 0.41–0.90], *p* = 0.014 and HR0.41 [95% CI 0.26–0.63], *p* < 0.001, respectively). SIPS predicted PFS (HR 1.60 [95% CI 1.34–1.90], *p* < 0.001) and OS (HR 1.69 [95% CI 1.41–2.02], *p* < 0.001). SIPS predicted the occurrence of any irAE in all patients (*p* = 0.011), but not on 24-week landmark analyses (*p* = 0.174). The occurrence of irAEs predicted favourable outcomes regardless of the baseline inflammatory status (*p* = 0.015). Conclusion: The occurrence of certain irAEs is associated with a survival benefit in patients with NSCLC expressing PD-L1 ≥ 50% receiving pembrolizumab. We find that the association between low levels of systemic inflammation and the risk of irAEs is confounded by their independent prognostic value.

## 1. Introduction

Immune checkpoint inhibitor (ICI) therapies have revolutionised the management of advanced or metastatic non-small-cell lung cancer (NSCLC). In particular, the anti-programmed cell death protein 1 (anti-PD-1) inhibitor pembrolizumab, either alone or in combination with cytotoxic chemotherapy, is established as a key component of first-line systemic anticancer therapy (SACT) [1,2,3,4]. A key benefit of ICI therapy is the potential for the durable control of disease. In the KEYNOTE-024 trial, patients with NSLC expressing PD-L1 ≥ 50% treated with pembrolizumab monotherapy experienced double the five-year survival rate compared to patients who received cytotoxic chemotherapy (32% v 16%, respectively) [1,2]. 

The potential for this benefit of ICI therapy is tempered by the risk of immune-related adverse events (irAEs) [1,2,5]. These adverse events, specific to immunotherapy treatments, occur in up to 66% of patients with NSCLC and may affect any organ, including the lungs, skin, gastrointestinal tract, liver, endocrine glands, and central nervous system [5,6]. They result from the heightened immune response, with PD-1 and its ligand (PD-L1) being critical molecules involved in T-cell regulation [7]. IrAEs may be clinically significant events, with high-grade events (i.e., National Cancer Institute Common Terminology Criteria for Adverse Events (CTCAE) v. 5.0 ≥ 3) frequently managed with immunosuppressive high-dose steroids and cessation of further treatment [5]. There is also the potential for life-changing, long-term toxicity or death. 

It has been established that the occurrence of irAEs predict enhanced ICI efficacy [8,9,10,11,12]. In patients with NSCLC treated with ICI, irAEs were associated with a higher objective response rate (OOR) (risk ratio: 2.43 (95% CI 2.06–2.88), *p* < 0.00001), improved progression-free survival (PFS) (HR0.50 (95% CI: 0.44–0.57), *p* < 0.00001), and overall survival (OS) (HR0.51 (0.43–0.61)) [13]. Subgroups of patients with irAEs, including those relating to dermatological, endocrine, and gastrointestinal toxicity, demonstrated significantly better OS. A possible explanation for this is that these toxicities are less severe, or are less likely to be life-threatening. By contrast, hepatobiliary, pulmonary, and high-grade irAEs were not correlated with survival, with these toxicities more likely to be life-threatening [13]. 

A unifying predictive biomarker that describes the risk/reward benefit for ICI treatment in terms of survival and irAEs would be a valuable tool in the oncology clinic. It is widely established that the inflammatory status of the patient predicts survival, quality of life, and treatment efficacy across all solid tumours including NSCLC [14,15,16]. Taking this further, we have also demonstrated that biomarkers of systemic inflammation (Scottish Inflammatory Prognostic Score [SIPS]; combining albumin and neutrophil count [NC]) predict survival in patients with NSCLC treated with pembrolizumab [17]. We hypothesise that the inflammatory status of the patient may be related to irAEs and/or survival. Therefore, the aim of the present study was to examine how the inflammatory status may be related to irAEs and/or survival in patients with NSCLC expressing PD-L1 ≥ 50% who received first-line pembrolizumab monotherapy. 

## 2. Methods

### 2.1. Study Population

All patients being treated with first-line pembrolizumab monotherapy for advanced NSCLC in a large regional cancer centre in Scotland, UK between June 2016 and April 2022 were identified from the South East Scotland Cancer Network Lung Cancer Database. Eligible patients were 18 years or over; had a pathological diagnosis of NSCLC, PDL1 expression ≥ 50%, with no sensitising EGFR, ROS1, or ALK molecular aberrations; and had received at least one dose of pembrolizumab therapy. 

### 2.2. Procedure and Assessment

Patient demographics, clinicopathological characteristics, and irAE incidences were recorded. Eastern Co-operative Oncology Group Performance Status (PS), white cell count (WCC), NC, lymphocyte count (LC), platelets, and albumin within 14 days prior to cycle 1, day 1 pembrolizumab monotherapy were recorded. Neutrophil/lymphocyte ratio (NLR) (NC (×10^9^/L)/LC (×10^9^/L)), platelet/lymphocyte ratio (PLR) (platelets (×10^9^/L)/LC (×10^9^/L)), prognostic nutritional index (PNI) (albumin (g/L) + (5 × LC (×10^9^/L)), and SIPS (1 point each for albumin < 35 g/L and NC > 7.5 × 10^9^/L to give a 3-tier categorical score) were calculated. WCC, NC, and albumin were categorised within normal limits, in line with previous work in this area [17,18,19,20]. Cut-offs for NLR (≤/>5) [21,22], PLR (≤/>180) [23,24], and PNI (≥/<45) [25,26] were based on previous studies examining these factors and not derived from the present analysis [17]. 

The diagnosis of irAEs followed internationally recognised guidelines [5]. Grading of irAEs was determined using the National Cancer Institute Common Terminology Criteria for Adverse Events v. 5.0 [27]. Patients were monitored for the development of irAEs per routine clinical practice, with clinical review at each visit including regular examination, and laboratory and radiological assessments. irAEs were grouped into mild (i.e., CTCAE grade 1–2) or severe (i.e., CTCAE grade ≥ 3). Treatment with steroids was defined as any steroid use ≥ 10 mg oral prednisolone equivalent by any route. Patients were considered to have discontinued pembrolizumab secondary to irAEs if this decision was documented in the patient clinical record. 

Progression-free survival (PFS), defined as the number of months from C1D1 pembrolizumab until radiological or clinical evidence of progressive disease prompting cessation of treatment, or censorship (11 May 2023) if there is no evidence of progressive disease at follow-up date, was calculated. Overall survival (OS), defined as the number of months from C1D1 pembrolizumab until death, or censorship (11 May 2023) if alive at follow-up date, was calculated. 

Univariate analysis of survival and calculation of hazard ratios was performed using Cox’s proportional hazards model. Multivariate analysis of survival was carried out using a backward conditional approach: variables with a *p*-value > 0.10 were removed in a stepwise fashion to leave only those with an independent significant relationship with survival. The Kaplan–Meier method was used to plot survival curves. Log-rank testing and Mantel–Byar time-dependent analysis were applied to assess statistically significant differences in survival [28]. Odds ratios and Student’s *t*-test were used to assess associations between variables. 

## 3. Results

Table 1 details the patient characteristics. Two hundred and sixty-two patients were eligible. Of these, 56% were female and the median age of the cohort was 68 years (interquartile range (IQR) of 62–73 years).

The median PFS was 5.9 (IQR 1.6–20.8) months. Forty-five (17%) patients had no evidence of progressive disease at censorship, in whom the minimum and median follow-up was 12.4 and 35.2 months, respectively. The median OS was 11.70 (IQR 3.6–33.2) months. Fifty-nine (23%) patients were alive at censorship, in whom the minimum and median follow-up was 12.4 and 33.7 months, respectively. Eighty-three (32%) patients experienced an irAE. There were no significant differences between those who did and did not experience an irAE in terms of routine clinicopathological features. However, a trend towards a higher incidence of irAEs was observed in patients aged ≥ 75 year (*p* = 0.086) and those with a PD-L1 expression ≥ 90% (*p* = 0.055). 

Figure 1 shows the Kaplan–Meier survival analyses according to irAE status. The occurrence of any irAEs was associated with prolonged PFS (HR 0.41 (95% CI 0.30–0.56), *p* < 0.001) and OS (HR 0.36 (95% CI 0.26–0.50), *p* < 0.001) (Figure 1a). These associations were maintained at both the 12-week (Figure 1b) and 24-week (Figure 1c) landmark analyses. The occurrence of any irAE was also associated with prolonged OS (HR0.53 (95% CI 0.78–0.73), *p* < 0.001), but not PFS (HR0.77 (95% CI 0.57–1.05), *p* = 0.095) on the Mantel–Byar time-dependent analyses. 

Table 2 provides detailed clinical information for individual irAEs by the organ affected. One hundred and two irAEs were recorded, with 17 (6%) patients experiencing > 1 irAE. The majority (69%) of irAEs were mild (i.e., grade 1–2). The most frequent irAEs were endocrine (n = 33 (32%)), dermatological (n = 19 (19%)), and gastrointestinal (n = 13 (13%)). Gastrointestinal, pulmonary, and hepatic irAEs were most frequently severe (62%, 80%, and 64%, respectively). One patient died as a result of irAE pneumonitis (treatment-related mortality rate of 0.4%).

Thirty-one (12%) patients experienced a severe irAE and 52 (20%) experienced only mild irAEs. All patients with severe irAEs received oral steroids, as did 22 patients with mild irAEs (total n = 53 (64%)). All patients with severe irAEs discontinued pembrolizumab, as did 12 patients with mild irAEs (total n = 44 (43%)). 

The median time to first irAE was 3.4 (IQR 1.4–7.4) months. Forty-three (42%) and sixty-eight (67%) irAEs occurred within 12 and 24 weeks of starting pembrolizumab, respectively. No significant difference in the time to mild or severe irAEs was observed (3.4 (IQR 2.0–7.5) months v 3.4 (0.7–8.3) months, respectively, *p* = 0.533). 

Dermatological (*p* = 0.002), endocrine (*p* < 0.001), gastrointestinal (*p* < 0.001), hepatic (*p* = 0.0.11), and musculoskeletal (*p* = 0.039) irAEs were associated with prolonged PFS compared to the absence of irAEs. Dermatological (*p* = 0.001), endocrine (*p* < 0.001), gastrointestinal (*p* = 0.001), and hepatic (*p* = 0.005) irAEs were associated with prolonged OS compared to the absence of irAEs (Appendix A). Patients with pulmonary irAEs demonstrated PFS and OS most similar to patients who did not experience irAEs. 

Table 3 shows the relationship between irAEs and survival. Severe irAEs were associated with improved PFS versus no irAEs in all patients (*p* = 0.024), but this association was lost at the 12-week (*p* = 0.094) and 24-week landmark (Appendix A). No association was seen between severe irAEs and PFS on the Mantel–Byar time-dependent analyses (HR1.04 (95% CI 0.91–1.20), *p* = 0.567). Severe irAEs were also associated with improved OS versus no irAEs in all patients (*p* = 0.008) and at the 12-week landmark (*p* = 0.011), but the association was lost at the 24-week landmark (*p* = 0.074) (Appendix A). Again, there was no association between severe irAEs and OS on the Mantel–Byar time-dependent analyses (HR0.92 (95% CI 0.79–1.06), *p* = 0.258). By comparison, mild irAEs were associated with improved PFS and OS in all landmark analyses (Appendix A). On the Mantel–Byar time-dependent analyses, mild irAEs were associated with both improved PFS (HR0.61 (95% CI 0.41–0.90), *p* = 0.014) and OS (HR0.41 (95% CI 0.26–0.63), *p* < 0.001).

Amongst patients with any irAE, those who received steroids had poorer OS than those who did not receive steroids (HR 1.95 (95% CI 1.04–3.67), *p* = 0.037), but no significant differences between PFS was observed (Appendix A). When only patients with mild irAEs were considered, steroid use was not associated with differences in PFS or OS. When treatment discontinuation due to irAEs was considered as a variable, patients with any irAE who discontinued treatment had poor PFS (HR 1.76 (95% CI 1.03–3.01), *p* = 0.041) and OS (HR 2.00 (95% CI 1.10–3.65), *p* = 0.024), but, again, these differences were not observed with respect to mild irAEs (Appendix A). Amongst 17 patients who experienced multiple irAEs, 12 (71%) experienced at least one endocrine irAE. No statistically significant differences in survival were observed between patients with single or multiple irAEs. 

As we have previously demonstrated, biomarkers of systemic inflammation at pre-treatment baseline were predictive of survival (Appendix A). On multivariate analyses, SIPS was the only inflammatory biomarker to predict PFS (HR1.64 (95% CI 1.38–1.96), *p* < 0.001), stratifying PFS from 1.6 months (SIPS2) to 7.1 months (SIPS1) to 9.9 months (SIPS0) (*p* < 0.001). SIPS remained predictive of PFS in the 12-week (HR 1.43 (95% CI 1.13–1.81), *p* = 0.003) and 24-week (HR 1.42 (95% CI 1.06–1.91), *p* = 0.018) landmark cohorts (Appendix A. SIPS was also predictive of OS on the multivariate analysis (HR 1.69 (95% CI 1.41–2.02), *p* < 0.001), stratifying OS from 3.1 months (SIPS 2) to 12.4 months (SIPS1) to 19.9 months (SIPS0). Again, it remained predictive of OS in the 12-week (HR 1.56 (95% CI 1.26–1.94), *p* < 0.001) and 24-week (HR 1.49 (95% CI 1.16–1.89), *p* = 0.001) landmark cohorts (Appendix A). In addition to SIPS, on multivariate analyses, PS was independently predictive of PFS (HR1.40 (95% CI 1.07–1.83), *p* = 0.012) and OS (HR 1.49 (95% CI 1.12–1.98), *p* = 0.006) (Appendix A). PD-L1 status was also independently predictive of PFS (HR 0.71 (0.54–0.93), *p* = 0.012), but no significant associations were seen between PD-L1 status and OS (Appendix A). 

Table 4 describes the association between pre-treatment baseline biomarkers of systemic inflammation status and the occurrence of iRAEs. Lower levels of systemic inflammation as defined by WCC, NC, albumin, and SIPS were associated with a significantly higher risk of irAEs. However, no association between pre-treatment baseline WCC, NC, albumin, or SIPS status and the occurrence of irAEs was observed in the 12- and 24-week landmark cohorts (*p* > 0.05). NLR, PLR, and PNI did not predict the incidence of irAEs in our cohort.

Finally, we examined the prognostic significance of WCC, NC, albumin, or SIPS and the occurrence of irAEs in combination (Table 5). We found that, regardless of the level of baseline systemic inflammation, as defined by WCC, NC, albumin, and SIPS, patients who experienced irAEs demonstrated more favourable PFS and OS. These same biomarkers of systemic inflammation were predictive of PFS and OS in patients who did not experience an irAE. In patients who experienced an irAE, neither WCC, NC, albumin, or SIPS were predictive of PFS, but SIPS and albumin stratified OS from 16.7 months (SIPS 2), to 23.6 months (SIPS 1), to 34.3 months (SIPS 0) (*p* = 0.037), and from 16.7 months (albumin < 35 g/L) to 43.9 months (albumin ≥ 35 g/L) (*p* = 0.005), respectively.

## 4. Discussion

This study reports the association between irAEs, biomarkers of systemic inflammation, and clinical outcomes in patients with NSCLC expressing PDL1 ≥ 50% treated with pembrolizumab. The work benefits from a large, well-characterised, and uniform population, with the inclusion of time-dependent survival analyses. We confirm that the occurrence of irAEs is associated with survival, but found that these differences are not observed in all subgroups of irAEs. We also provide further validation that biomarkers of systemic inflammation predict survival in this patient group. Although we identify that there is a higher risk of irAEs in patients with lower levels of baseline pre-treatment systemic inflammation, we find that these associations are lost when time-dependent analyses are performed. 

Our findings closely mirror those of recent large meta-analyses which demonstrated that the occurrence of any irAE was a strong predictor of efficacy in patients with NSCLC treated with ICIs [13,29]. We found similar differences in the prognostic significance of specific irAEs. Although both PFS and OS were significantly longer in patients experiencing dermatological, endocrine, and gastroenterological irAEs, no significant difference was observed in patients with pulmonary irAEs. The latter finding is unsurprising given the respiratory frailty of patients with NSCLC [30]. Contrary to previous findings, hepatic irAEs were associated with more favourable survival in our cohort [13,29]. 

Immortal time bias (ITB) is common in observational studies and describes the situation when groups of participants cannot experience an outcome of interest during some period of the follow-up [28,31,32]. In this study, patients with shorter survival have a lower chance of irAE development due to the shorter duration of ICI exposure. Not all studies have taken ITB into account when assessing the association between irAEs and survival. In those that have, landmark analyses have most frequently been used [9,12,29,30,33,34,35,36]. We demonstrate that the positive association between any irAE occurrence and survival is maintained on landmark analyses, suggesting this is a true effect. A limitation of this approach, though, is that results may differ depending on the choice of landmark used. In this example, a short interval may continue to include patients with only a limited exposure to treatment. To further validate our findings, we performed time-dependent analyses using the Mantel–Byar method. In this approach, every patient contributes both unexposed and exposed patient time according to their individual exposure status during follow-up. It is felt to be a more conservative model, with the potential to underestimate survival in the exposed cohort (i.e., with irAE) [37]. In our study, it provides additional reassurance that the observed impact of any irAE on OS is a true association. To our knowledge, this is one of only a handful of studies to have applied such a robust methodology [9].

Significantly, the survival benefit associated with the occurrence of irAEs in our cohort is restricted to mild irAEs. In a recent study of patients with NSCLC receiving chemoimmunotherapy, an approach typically restricted to those with PDL1 expression < 50% in our practice, patients with mild irAEs demonstrated improved PFS and OS compared to patients with severe or no irAEs [8]. That study benefited from the use of landmark analyses at 12 and 24 weeks, in which the development of mild irAEs remained predictive of OS, but not PFS. Our data support these findings, with patients experiencing mild irAEs having more favourable survival outcomes than those without irAEs at each landmark and on time-dependent analyses. Although severe irAEs were associated with improved OS in all patients and at 12 weeks, this association was lost in the 24-week landmark cohort. We note that only one additional patient was excluded from the severe irAE cohort between the 12- and 24-week landmarks, suggesting this effect was not due to significantly smaller sample sizes. Indeed, there was no association between OS and severe irAEs on time-dependent analyses. 

Several reasons for the observed differences in clinical benefit between mild and severe irAEs have been proposed. Mild irAEs may be detected sub-clinically (e.g., by routine thyroid hormone monitoring), whereas severe irAEs typically cause symptoms and may be life-threatening themselves [5]. Although, in our cohort, only one patient’s death was directly attributable to an irAE, the sequelae of severe irAEs and the need for immunosuppression may contribute to other events that lead to death not detected in our dataset. Steroid use appears unfavourable if it is required for cancer-related indications or is initiated in the early part of ICI treatment, but not if it is required for irAE management, as we find here [38,39,40]. Severe irAEs invariably result in the cessation of further ICI therapy [5]. Feasibly, this may result in poorer outcomes as these patients have less exposure to ICI. However, this remains unclear in our cohort. We note that the majority of irAE events in patients with more favourable dermatological or endocrine irAEs did not require oral steroids and patients were able to continue with treatment. However, patients with gastrointestinal or hepatic irAEs had favourable outcomes, but events were predominately severe, all patients received steroids, and the majority discontinued treatment. Further work is needed to understand these differences.

The identification of factors that predict the likelihood of irAEs would be a useful tool in the clinic when counselling patients on the risks and benefits of ICI therapy. Clinicopathological factors such as age, performance status, active or former smoking status, and a higher PD-L1 expression may be associated with an increased risk of irAEs [41,42]. Patients with pre-existing auto-immune diseases, particularly those requiring immunosuppressive therapy, are typically excluded from ICI therapy due to the high risk of irAE development [1,2,3,4]. In our practice, pre-existing pulmonary fibrosis or ground glass attenuation on CT imaging is also considered contraindication to ICI, with studies suggesting this is a risk factor for pneumonitis in patients with NSCLC [43]. Laboratory markers of systemic inflammation have also been suggested as biomarkers of irAE risk. Amongst these, NLR has been most extensively investigated, albeit with conflicting results. Although several studies have shown low NLR to be a predictor for developing irAE, others have found no association [36,42,44,45].

As we have previously demonstrated in this setting, biomarkers of systemic inflammation are predictive of survival in patients with NSCLC expressing PD-L1 ≥ 50% treated with pembrolizumab [17]. Most studies have focused on pre-treatment or baseline measurements of biomarkers of systemic inflammation with respect to any irAE risk. We find that low levels of baseline systemic inflammation as evidenced by WCC, NC, albumin, and SIPS, but not NLR, PLR, or PNI, are predictive of any irAE. However, in the 12-week landmark cohort, only albumin and SIPS remained predictive of any irAE, with these associations subsequently lost in the 24-week landmark cohort. A possible explanation for this observation, returning to the issue of ITB, is that patients with higher levels of inflammation have a poorer prognosis, and, hence, less opportunity to develop an irAE. In support of this, we note that the 12- and 24-week landmark cohorts were less inflamed by each baseline biomarker than the entire cohort. This suggests that the individual prognostic value of irAEs and biomarkers of systemic inflammation is a confounder in associations between these two factors. We, therefore, urge caution when attributing associations between biomarkers of systemic inflammation and the risk of any irAEs. 

WCC, NC, albumin, and SIPS stratified PFS and OS in patients without irAE; however, in patients experiencing any irAE, the associations were less consistent. Significantly though, the occurrence of any irAE was associated with improved survival regardless of baseline systemic inflammation status. Survival outcomes for patients with high inflammation and irAEs were the same or better than those observed in patients with low inflammation without irAEs. In an unselected cohort of patients treated with anti-PD1 monotherapy, NLR was raised at the time of irAE occurrence and was significantly elevated up to 4 weeks before the diagnosis of pneumonitis [46]. The continuous monitoring of NLR identified patients with a subsequent prompt reduction in NLR after irAEs who have more favourable PFS and OS compared to patients who maintained a high NLR. This suggests an interplay between irAEs and systemic inflammation. Significantly, we have previously demonstrated that biomarkers of systemic inflammation are predictive of subsequent survival whenever they are measured during treatment [47]. A key question here is whether changes in inflammatory status reflect irAEs, their management with immunosuppressive treatments, or the effects of diminished cancer activity in response to therapy.

In addition to biomarkers of systemic inflammation, PS independently predicted both PFS and OS. PS is recognised as a gold-standard prognostic factor for patients with cancer. Previous studies have identified a very high PD-L1 score (i.e., ≥90%) as being predictive of survival outcomes in patients with NSCLC treated with first-line pembrolizumab [48,49]. In our cohort, PD-L1 status was also predictive of PFS on multivariate analyses, but, although a trend was seen towards improved OS on univariate analyses, this finding was not significant. It is interesting that we observed a non-significant trend towards a higher incidence of irAEs in patients with a very high PD-L1 expression. Further work is required to understand the association between PD-L1 status, irAE risk, and survival. 

Understanding how we can apply these findings is an unmet need in clinical practice. We continue to advocate for the use of prognostic biomarkers of systemic inflammation in discussions with patients about the expected benefits of treatment. However, we surmise that these same biomarkers tell us little about the risk of harm associated with ICI irAEs. It is widely accepted that the early detection and prompt management of irAEs are critical for avoiding substantial harm to patients [5]. In the absence of a proven sensitive and specific biomarker, strategies such as improved patient education and the standardisation of irAE surveillance have been proposed. Presumably, better early detection and treatment of irAEs would mean fewer patients progress to severe irAEs. Modifications to treatment regimens may also reduce the risk of irAEs. Much of this work has focused on a combination of ipilimumab and nivolumab, where both the dose and scheduling of ipilimumab correlates with safety and tolerability [50]. Studies such as REFINE-lung, REFINE-renal/melanoma, and PRISM are exploring whether the ICI dose frequency de-escalation of ICI remains efficacious and is safer with respect to irAEs [51,52,53]. We advocate for the inclusion of biomarkers of systemic inflammation within this work.

A limitation of this study is the inclusion of patients from a single centre, including a subset we have previously investigated. However, our cohort is well-characterised and is larger than many which have explored the association of biomarkers of systemic inflammation or irAEs and survival outcomes. We would support the validation of these findings in a large prospective multicentre study. This is feasible given the use of standardised treatment pathways, the grading of irAE severity, and the ready availability of biomarkers of systemic inflammation around the world. Other biomarkers of systemic inflammation, such as the modified Glasgow Prognostic Score (mGPS), have been shown to be predictive of survival and the occurrence of irAEs in NSCLC treated with pembrolizumab. However, our cohort lacked c-Reactive Protein (CRP) measurements as previously described [17]. We also note that SIPS has not yet been externally validated, but the other biomarkers of systemic inflammation reported here have been the subject of numerous studies.

## 5. Conclusions

The results of this study demonstrate that any irAEs are predictive of survival in patients with NSCLC expressing PDL1 ≥ 50% treated with pembrolizumab, but that these associations are restricted to subsets of irAEs. Most significantly, whilst patients with mild irAEs experience better clinical outcomes, those with severe irAEs demonstrate no improvement in survival. Although biomarkers of systemic inflammation have previously been identified as potential predictors of irAE risk, we find that these associations are confounded by their independent prognostic significance. The development of irAEs improves survival regardless of the baseline pre-treatment prognostic biomarkers of inflammation status. We continue to support the use of this objective prognostic information in open discussions with patients regarding the potential benefits of pembrolizumab therapy. 

## Figures and Tables

**Figure 1 cancers-15-05502-f001:**
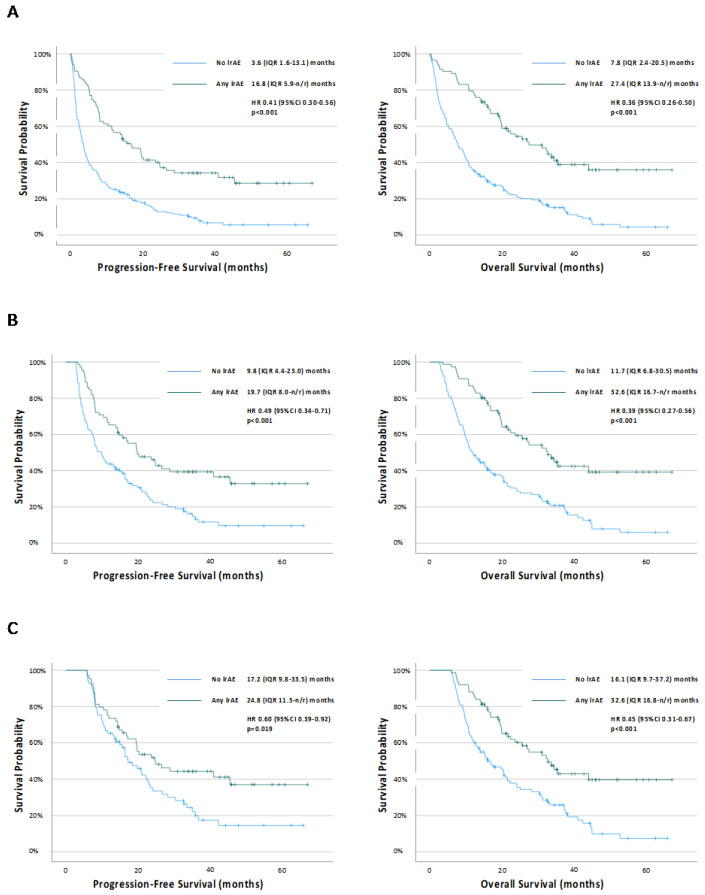
Kaplan–Meier survival curves examining the relationship between the occurrence of immune-related adverse events and progression-free survival or overall survival in (**A**) all patients, and (**B**) 12-week landmark and (**C**) 24-week landmark cohorts. n/r = not reached.

**Table 1 cancers-15-05502-t001:** Patient characteristics and the relationship between routine clinicopathological variables and incidence of immune-related adverse events in patients with NSCLC treated with first-line pembrolizumab (* non-squamous only, where available (*n* = 202)).

Characteristics		Total	Without irAE	With irAE	*p*
		n = 262	n = 179	n = 83	
		n (%)	n (%)	n (%)	
Age (years)	<75	220 (84)	155 (87)	65 (78)	0.089
	≥75	42 (16)	24 (13)	18 (22)	
Sex	Male	114 (44)	81 (45)	33 (40)	0.404
	Female	148 (56)	98 (55)	50 (60)	
ECOG-performance status	0/1	218 (83)	146 (82)	72 (87)	0.296
	2	44 (17)	33 (18)	11 (13)	
Histologic subtype	Squamous	54 (21)	32 (18)	22 (17)	0.108
	Non-squamous	208 (79)	147 (82)	61 (73)	
PD-L1 expression	<90	127 (48)	94 (53)	33 (40)	0.055
	≥90	135 (52)	85 (47)	50(60)	
KRAS status *	Wildtype	95 (47)	65 (45)	30 (51)	0.485
	Mutant	107 (53)	78 (55)	29 (49)	

**Table 2 cancers-15-05502-t002:** Detailed clinical information for patients with or without irAE and irAE events by the organ affected. n/a = not applicable. Bold/italics represent statistically significant findings (*p* < 0.005).

	Total	Mild irAE (Grade 1–2)	Severe irAE (Grade 3–5)	Time to iRAE (months)	Treated with Steroids	Pembrolizumab Discontinued Secondary to irAE	Progression-Free Survival	Overall Survival
	n (%)	n (%)	n (%)	Median (IQR)	n (%)	n (%)	HR (95% CI) *p*	HR (95% CI) *p*
Patients								
Without irAE	179 (68)	n/a	n/a	n/a	n/a	n/a	ref	ref
With irAE	83 (32)	52 (63)	31 (37)	3.4 (1.4–7.4)	53 (65)	42 (51)	** *0.41 (0.30–0.56)* ** ** *<0.001* **	** *0.36 (0.26–0.50) * ** ** *<0.001* **
irAE Events								
All	102	70 (69)	32 (31)	3.5 (2.0–7.5)	58 (57)	44 (43)	n/a	n/a
Dermatological	19 (19)	13 (68)	6 (19)	3.0 (1.4–7.9)	9 (47)	7 (37)	** *0.37 (0.20–0.69) * ** ** *0.002* **	** *0.35 (0.19–0.67) * ** ** *0.001* **
Endocrine	33 (32)	≥28 (≥85)	≤5 (≤15)	3.4 (2.0–12.3)	≤5 (≤15)	≤5 (≤15)	** *0.36 (0.23–0.57) * ** ** *<0.001* **	** *0.30 (0.18–0.49) * ** ** *<0.001* **
Gastrointestinal	13 (13)	≤5 (≤38)	≥8 (≥62)	7.3 (5.7–12.3)	13 (100)	11 (85)	** *0.23 (0.10–0.52) * ** ** *<0.001* **	** *0.21 (0.09–0.52) * ** ** *0.001* **
Hepatic	11 (11)	≤5 (≤45)	≥6 (≥55)	3.4 (1.2–6.7)	8 (73)	8 (73)	** *0.34 (0.15–0.78) * ** ** *0.011* **	** *0.27 (0.11–0.67) * ** ** *0.005* **
Musculoskeletal	8 (8)	8 (100)	0 (0)	2.3 (1.9–5.0)	8 (100)	≤5 (≤63)	** *0.42 (0.19–0.96) * ** ** *0.039* **	*0.49 (0.22–1.11) * *0.088*
Pulmonary	10 (10)	≤5 (≤50	≥5 (≥50)	2.3 (0.3–5.0)	10 (100)	9 (90)	*1.12 (0.57–2.18) * *0.752*	*1.33 (0.68–2.61) * *0.408*
Other (Cardiac, Neurological, Renal)	8 (8)	8 (100)	0 (0)	4.1 (1.7–6.6)	6 (75)	6 (75)	*n/a*	*n/a*

**Table 3 cancers-15-05502-t003:** Association between mild (i.e., grade 1–2) or severe (i.e., grade 3–5) immune-related adverse events and progression-free survival or overall survival in all patients, and 12-week landmark and 24-week landmark cohorts. n/r = not reached. Bold/italics represent statistically significant findings (*p* < 0.005).

	PFS	HR	*p*	OS	HR	*p*
All						
No irAE	3.6 (1.3–13.1)	ref	ref	7.8 (2.4–20.5)	ref	ref
Severe irAE	7.8 (2.2–20.2)	** *0.61 (0.40–0.94)* **	** *0.024* **	16.1 (7.3–34.3)	** *0.55 (0.35–0.86)* **	** *0.008* **
Mild irAE	24.7 (8.1-n/r)	** *0.32 (0.22–0.48)* **	** *<0.001* **	35.3 (19.7-n/r)	** *0.27 (0.18–0.42)* **	** *<0.001* **
12-week Landmark						
No irAE	9.8 (4.4–23.0)	ref	ref	11.7 (6.8–30.5)	ref	ref
Severe irAE	14.3 (7.0-n/r)	0.64 (0.38–1.08)	0.094	22.3 (13.9-n/r)	** *0.51 (0.31–0.86)* **	** *0.011* **
Mild irAE	24.8 (10.4-n/r)	** *0.43 (0.28–0.66)* **	** *<0.001* **	35.3 (19.8-n/r)	** *0.33 (0.21–0.52)* **	** *<0.001* **
24-week Landmark						
No irAE	17.2 (9.8–33.5)	ref	ref	16.1 (9.7–37.2)	ref	ref
Severe irAE	14.7 (7.8-n/r)	0.81 (0.44–1.47)	0.481	22.3 (13.9-n/r)	0.62 (0.37–1.05)	0.074
Mild irAE	28.8 (14.3-n/r)	** *0.51 (0.31–0.84)* **	** *0.008* **	43.9 (19.8-n/r)	** *0.38 (0.23–0.61)* **	** *<0.001* **

**Table 4 cancers-15-05502-t004:** The association between pre-treatment biomarkers of systemic inflammation and risk of immune-related adverse event in all patients, and 12-week landmark and 24-week landmark cohorts. Bold/italics represent statistically significant findings (*p* < 0.005).

Characteristics		All			12-Week Landmark			24-Week Landmark		
		Without irAE	With irAE	OR (95% CI) *p*	Without irAE	With irAE	OR (95% CI) *p*	Without irAE	With irAE	OR (95% CI) *p*
		n = 179	n = 83		n = 130	n = 76		n = 105	n = 75	
		n (%)	n (%)		n (%)	n (%)		n (%)	n (%)	
White Cell Count	≤11.0 × 10^9^/L	**109 (61)**	**64 (77)**	** *2.16* ** ** *(1.20–3.91)* ** ** *0.011* **	*89 (68)*	*59 (78)*	*1.60* *(0.83–3.08)* *0.158*	*80 (76)*	*59 (79)*	*1.15 * *(0.57–2.35)* *0.696*
	>11.0 × 10^9^/L	70 (39)	19 (23)		*41 (32)*	*17 (22)*		*25 (24)*	*16 (21)*	
	Median (IQR)	10.1 (7.8–13.2)	8.5 (6.8–10.8)	** *0.008* **	*9.5 (7.3–11.6)*	*8.4* *(6.8–10.6)*	*0.191*	*9.3* *(7.1–11.5)*	*8.4* *(6.8–10.6)*	*0.198*
Neutrophil Count	≤7.5 × 10^9^/L	101 (56)	63 (76)	** *2.43* ** ** *(1.35–4.36)* ** ** *0.003* **	*84 (65)*	*58 (76)*	*1.76 * *(0.93–3.34)* *0.082*	*70 (67)*	*58 (77)*	*1.71* *(0.87–3.35)* *0.120*
	>7.5 × 10^9^/L	78 (44)	20 (24)		*46 (35)*	*18 (24)*		*35 (33)*	*17 (23)*	
	Median (IQR)	6.8 (5.1–10.0)	5.9 (4.5–7.4)	** *0.010* **	*5.8* *(4.5–7.4)*	*6.1 (4.8–8.4)*	*0.245*	*5.7* *(4.6–7.9)*	*6.0* *(4.6–8.3)*	*0.624*
NLR	<5	104 (58)	51 (61)	*1.15* *(0.67–1.96)* *0.608*	*89 (68)*	*49 (64)*	*0.84 * *(0.46–1.52)* *0.557*	*75 (71)*	*49 (65)*	*0.76* *(0.40–1.43)* *0.384*
	≥5	75 (42)	32 (39)		*41 (32)*	*27 (36)*		*30 (29)*	*26 (35)*	
	Median (IQR)	4.3 (2.8–7.4)	4.3 (2.8–6.5)	*0.122*	*3.8 (2.6–2.0)*	*3.9 (2.8–5.7)*	*0.524*	*3.6* *(2.6–5.1)*	*4.4* *(2.9–6.6)*	*0.737*
PLR	≤180	51 (28)	28 (34)	*1.28* *(0.73–2.23)* *0.390*	*41 (32)*	*28 (37)*	*1.27 * *(0.70–2.30)* *0.437*	*39 (37)*	*28 (62)*	*1.01* *(0.55–1.86)* *0.979*
	>180	128 (72)	55 (66)		*89 (68)*	*48 (66)*		*66 (63)*	*47 (38)*	
	Median (IQR)	225 (169–321)	238 (155–238)	*0.548*	*219* *(165–306)*	*226* *(145–336)*	*0.477*	*202* *(150–276)*	*260* *(163–352)*	*0.216*
Albumin	≥35 g/L	86 (48)	55 (66)	** *2.12* ** ** *(1.24–3.65)* ** ** *0.006* **	*71 (55)*	*53 (70)*	** *1.91* ** ** *(1.05–3.49)* ** ** *0.034* **	*62 (59)*	*52 (69)*	*1.57 * *(0.84–2.93)* *0.159*
	<35 g/L	93 (52)	28 (34)		*59 (45)*	*23 (30)*		*43 (41)*	*23 (31)*	
	Median (IQR)	34 (29–38)	36 (32–41)	** *0.005* **	*35 (31–39)*	*37 (33–41)*	** *0.07* **	*35 (31–38)*	*36 (33–41)*	*0.285*
PNI	≥45	66 (37)	39 (47)	*1.52 * *(0.90–2.57)* *0.121*	*55 (42)*	*38 (50)*	*1.36* *(0.77–2.41)* *0.285*	*49 (47)*	*37 (49)*	*1.11 * *(0.61–2.01)* *0.724*
	<45	113 (63)	44 (53)		*75 (68)*	*38 (50)*		*56 (53)*	*38 (51)*	
	Median (IQR)	42.0 (36.5–47.5)	44.8 (40.3–49.0)	*0.411*	*43.0 (38.7–48.0)*	*45.0 (41.4–49.5)*	*0.679*	*43.2* *(37.9–46.3)*	*45.6* *(40.8–49.3)*	*0.743*
SIPS	0	65 (36)	44 (53)	* **1.98** * * **(1.67–3.35)** * * **0.011** *	*56 (43)*	*42 (55)*	** *1.63* ** ** *(0.92–2.88)* ** ** *0.09* **	*48 (46)*	*42 (56)*	*1.51 * *(0.84–2.74)* *0.174*
	1/2	114 (64)	39 (47)		*74 (57)*	*34 (45)*		*57 (54)*	*33 (44)*	

**Table 5 cancers-15-05502-t005:** The association between biomarkers of systemic inflammation combined with the occurrence of irAEs and progression-free survival and overall survival in all patients. Bold/italics represent statistically significant associations in this table and others.

		Progression-Free Survival			Overall Survival		
		Without irAE	With irAE	HR (95% CI) *p*	Without irAE	With irAE	HR (95% CI) *p*
		Median (IQR)	Median (IQR)		Median (IQR)	Median (IQR)	
SIPS	0	7.6 (2.7–23.3)	14.7 (5.3-n/r)	** *0.57 (0.36–0.91)* ** ** *0.015* **	14.0 (2.8–5.3)	34.3 (18.9-n/r)	** *0.40 (0.24–0.67)* ** ** *0.001* **
	1	3.6 (1.8–13.6)	19.5 (7.1–45.4)	** *0.38 (0.22–0.64)* ** ** *<0.001* **	8.2 (3.0–21.5)	23.6 (12.3-n/r)	** *0.40 (0.24–0.69)* ** ** *0.001* **
	2	1.6 (0.8–4.0)	8.1 (7.5–28.8)	** *0.26 (0.11–0.64)* ** ** *0.003* **	2.9 (1.5–9.2)	16.7 (7.5–33.2)	** *0.35 (0.16–0.81)* ** ** *0.013* **
	HR (95% CI) *p*	** *1.75 (1.43–2.14)* ** ** *<0.001* **	1.15 (0.78–1.69) *0.479*		** *1.67 (1.37–2.04)* ** ** *<0.001* **	** *1.53 (1.03–2.28)* ** ** *0.037* **	
White Cell Count	≤11.0 × 10^9^/L	4.6 (1.8–16.4)	15.6 (5.9–45.4)	**0.50 (0.35–0.72)** **<0.001**	10.1 (4.1–28.2)	32.2 (14.3-n/r)	** *0.43 (0.29–0.63)* ** ** *<0.001* **
	>11.0 × 10^9^/L	1.6 (0.9–6.0)	19.6 (7.5-n/r)	**0.27 (0.14–0.51)** **<0.001**	4.1 (1.7–11.8)	25.5 (7.5-n/r)	** *0.28 (0.14–0.53)* ** ** *<0.001* **
	HR (95% CI) *p*	** *1.74 (1.26–3.40)* ** ** *0.001* **	0.85 (0.45–1.62) *0.621*		** *1.80 (1.30–2.49)* ** ** *<0.001* **	1.05 (0.53–2.07) 0.896	
Neutrophil Count	≤7.5 × 10^9^/L	11.0 (6.0–14.9)	20.4 (15.6–27.1)	** *0.54 (0.38–0.78)* ** ** *0.001* **	10.7 (4.5–31.0)	32.2 (14.3-n/r)	** *0.46 (0.31–0.68)* ** ** *<0.001* **
	>7.5 × 10^9^/L	3.3 (1.6–5.2)	17.9 (19.6–28.8)	** *0.25 (0.13–0.47)* ** ** *<0.001* **	4.1 (1.6–10.4)	25.5 (7.5-n/r)	** *0.26 (0.13–0.49)* ** ** *<0.001* **
	HR (95% CI) *p*	** *2.11 (1.54–2.91)* ** ** *<0.001* **	0.83 (0.44–1.57) *0.560*		** *2.02 (1.46–2.78)* ** ** *<0.001* **	1.06 (0.54–2.09) 0.868	
Albumin	≥35 g/L	6.0 (1.7–23.0)	17.1 (5.9-n/r)	** *0.48 (0.32–0.73)* ** ** *0.001* **	11.0 (4.6–32.6)	43.9 (19.2-n/r)	** *0.35 (0.22–0.56)* ** ** *<0.001* **
	<35 g/L	2.5 (1.2–5.9)	14.3 (5.9–26.5)	** *0.35 (0.22–0.57)* ** ** *<0.001* **	4.7 (1.7–11.7)	16.7 (8.0–32.6)	** *0.42 (0.26–0.67)* ** ** *<0.001* **
	HR (95% CI) *p*	** *1.97 (1.43–2.72)* ** ** *<0.001* **	1.53 (0.89–2.61) *0.123*		** *1.88 (1.37–2.59)* ** ** *<0.001* **	** *2.30 (1.29–4.10)* ** ** *0.005* **	

## Data Availability

Research data are available on reasonable request to the corresponding author.

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
