# Peer review of "Immune-Related Adverse Events, Biomarkers of Systemic Inflammation, and Survival Outcomes in Patients Receiving Pembrolizumab for Non-Small-Cell Lung Cancer"

_cancers, 2023, doi:10.3390/cancers15235502_

Round 1

Reviewer 1 Report

Comments and Suggestions for Authors

The manuscript by G. Ranyes and co-authors tries to answer a clinical very relevant question in the field of pembrolizumab in NSCLC: whether efficacy and/or immune related adverse events (irAE) can be predicted by clinical and/or laboratory markers from the blood?

The manuscript is written clearly. The presented data are concise and results are obtained and presented in an understandable way. Results are discussed properly.

There are some minor points:

1. In the methods part it should be described how precisely irAE were analyzed. Was there a questionnaire for patients and/or clinicians checking for CTC grades at every visit? Was it "only" laboratory findings and clinical examination? 

2. There seems to be a trend for increased irAE with PD-L1 very high TPS scores >90%.  As PD-L1 is to date the only prognostic factor in NSCLCL, these data should at least be discussed. In addition, Kaplan-Meier curves with PD-L1 50-90% and >90% for PFS and OS should be given (suppl figures).

3. SIPS predicted PFS and OS were clearly and was of independent prognostic significance in multivariate analysis. Therefore, SIPS 0-2 should be given as Kaplan-Meier curves for PFS and OS. 

4. Differences in PFS and OS in respect to different irAE are an important point, especially for clinicians. However, table 2 is overcrowded with information and difficult to read. Therefore, to make it more clear, an additional figure with bar graphs should be created with column pairs (no irAE versus irAE) with different irAE, e.g. dermatological, endocrine, pulmonary, hepatic. On the Y-axis the survival in months (PFS and OS) should be applied with asterisks indicating whether significant or not. With this figure the last two rows in table 2 may be futile and the figure can be read more clearly, especially when turned into landscape format.

5. Why was C-reactive protein (CRP) not analyzed in this study as a marker for systemic inflammation? There are data supporting that CRP might be a prognostic/predictive factor for immunotherapy in NSCLC.

6. If should be changed into is in line 361.

Author Response

The manuscript by G. Ranyes and co-authors tries to answer a clinical very relevant question in the field of pembrolizumab in NSCLC: whether efficacy and/or immune related adverse events (irAE) can be predicted by clinical and/or laboratory markers from the blood?

The manuscript is written clearly. The presented data are concise and results are obtained and presented in an understandable way. Results are discussed properly.

On behalf of all the authors, we thank reviewer 1 for their time in considering this manuscript and for the constructive feedback provided. We have taken these comments into account in the updated manuscript and provide feedback on each point below.

There are some minor points:

  1. In the methods part it should be described how precisely irAE were analyzed. Was there a questionnaire for patients and/or clinicians checking for CTC grades at every visit? Was it "only" laboratory findings and clinical examination? 

Point taken. This has been made clearer in the methods section (line 111-113):

The diagnosis of irAE followed internationally recognised guidelines (5). Grading of irAE was determined using the National Cancer Institute Common Terminology Criteria for Adverse Events v.5.0 (27). Patients were monitored for the development of irAE as per routine clinical practice, with clinical review at each visit including regular examination, laboratory and radiological assessments. irAEs were grouped into mild (i.e. CTCAE grade 1-2) or severe (i.e. CTCAE grade ≥3). Treatment with steroids was defined as any steroid use ≥10mg oral prednisolone equivalent by any route. Patients were considered to have discontinued pembrolizumab secondary to irAE if this decision was documented in the patient clinical record. 

  1. There seems to be a trend for increased irAE with PD-L1 very high TPS scores >90%.  As PD-L1 is to date the only prognostic factor in NSCLCL, these data should at least be discussed. In addition, Kaplan-Meier curves with PD-L1 50-90% and >90% for PFS and OS should be given (suppl figures).

Point taken. These comments have been taken into account:

Supplementary table 1 – PD-L1 status has been added to univariate and multivariate analyses of prognostic factors, with additional Kaplan Meier survival curves for PS and PD-L1 status added to the supplementary file.

Results - As we have previously demonstrated, biomarkers of systemic inflammation at pre-treatment baseline were predictive of survival (Supplementary Table 2). On mul-tivariate analyses SIPS was the only inflammatory biomarker to predict PFS (HR1.64 (95% CI 1.38-1.96), p<0.001), stratifying PFS from 1.6 months (SIPS2) to 7.1 months (SIPS1) to 9.9 months (SIPS0) (p<0.001). SIPS remained predictive of PFS in the 12-week (HR 1.43 (95%CI 1.13-1.81), p=0.003) and 24-week (HR 1.42 (95%CI 1.06-1.91), p=0.018) landmark cohorts (Supplementary Figure 6). SIPS was also predictive of OS on multivariate analysis (HR 1.69 (95% CI 1.41-2.02), p<0.001), stratifying OS from 3.1 months (SIPS 2) to 12.4 months (SIPS1) to 19.9 months (SIPS0). Again, it remained predictive of OS in the 12-week (HR 1.56 (95% CI 1.26-1.94), p<0.001) and 24-week (HR 1.49 (95%CI 1.16-1.89), p=0.001) landmark cohorts (Supplementary Figure 6). In addition to SIPS, on multivariate analyses PS was independently predictive of PFS (HR1.40 (95%CI 1.07-1.83), p=0.012) and OS (HR 1.49 (95%CI 1.12-1.98), p=0.006) (Supplementary Figure 7). PD-L1 status was also inde-pendently predictive of PFS (HR 0.71 (0.54-0.93), p=0.012), but no significant associations were seen between PD-L1 status and OS (Supplementary Figure 8).

Discussion - In addition to biomarkers of systemic inflammation, PS independently predicted both PFS and OS. PS is recognised as a gold-standard prognostic factor for patients with cancer and is routinely used in the assessment of patients. Previous studies have identified a very high PD-L1 score (i.e ≥90%) as being predictive of survival outcomes in patients with NSCLC treated with first-line pembrolizumab. In our cohort PD-L1 status was also predictive of PFS on multivariate analyses, but, although a trend was seen towards improved OS on univariate analyses, this finding was not significant. It is interesting that we observed a non-significant trend towards a higher incidence of irAEs in patients with a very high PD-L1 expression. Further work is required to understand the association between PD-L1 status, irAE risk and survival.

  1. SIPS predicted PFS and OS were clearly and was of independent prognostic significance in multivariate analysis. Therefore, SIPS 0-2 should be given as Kaplan-Meier curves for PFS and OS. 

Point taken. These curves have been added to the supplementary file.

  1. Differences in PFS and OS in respect to different irAE are an important point, especially for clinicians. However, table 2 is overcrowded with information and difficult to read. Therefore, to make it more clear, an additional figure with bar graphs should be created with column pairs (no irAE versus irAE) with different irAE, e.g. dermatological, endocrine, pulmonary, hepatic. On the Y-axis the survival in months (PFS and OS) should be applied with asterisks indicating whether significant or not. With this figure the last two rows in table 2 may be futile and the figure can be read more clearly, especially when turned into landscape format.

Point taken. The authors agree that table 2 is difficult to follow. We have considered the request to add a bar chart to show the differences in median PFS and OS. However, in practice we note that median survival was not reached for several irAE types, resulting in the suggested figure also being difficult to follow. Instead, we have rationalised the table by moving the PFS and OS “median (IQR)” columns to supplementary. Along with formatting changes, we feel that this makes the table clearer and better highlights the differences in survival as highlighted by cox regression analyses.

  1. Why was C-reactive protein (CRP) not analyzed in this study as a marker for systemic inflammation? There are data supporting that CRP might be a prognostic/predictive factor for immunotherapy in NSCLC.

 Point taken. This has been updated in the manuscript.

A limitation of this study if the inclusion of patients from a single centre, including a subset we have previously investigated. However, our cohort is well characterised and is larger than many which have explored the association of biomarkers of systemic in-flammation or irAE and survival outcomes. We would support validation of these findings in a large prospective multicentre study. This is feasible given the use of standardised treatment pathways, grading of irAE severity and ready availability of biomarkers of systemic inflammation around the world. Other biomarkers of systemic inflammation, such as the modified Glasgow Prognostic Score (mGPS), have been shown to be predictive of survival and the occurrence of irAE in NSCLC treated with pembrolizumab. However, our cohort lacked c-Reactive Protein (CRP) measurements as previously described (17). We also note that SIPS has not yet been externally validated, but the other biomarkers of systemic inflammation reported here have been the subject of numerous studies.

  1. If should be changed into is in line 361.

Point taken. This has been amended in the manuscript.

Reviewer 2 Report

Comments and Suggestions for Authors

The study by Raynes et al aims to examine how the inflammatory status may be related to irAE and/or survival in patients with NSCLC expressing PD-L1 ≥50% who received first-line pembrolizumab monotherapy. Authors report several interesting finding including associations between prolonged PFS and OS and dermatological, endocrine, gastrointestinal and hepatic, but not pulmonary, irAEs. Mild irAEs were associated 37 with better PFS and OS in all patients, including on time-dependent analyses. Also, the occurrence of irAEs predicted favorable outcomes regardless of baseline inflammatory status. Finally, authors report that the individual prognostic value of irAE and biomarkers of systemic inflammation is a confounder in associations between these two factors. The manuscript is well powered and well written. This reviewer has no major concern with publication of this manuscript.

Minor points/concern:

·         Did authors find any correlation / significant overlap between types of irAEs for example, gastrointestinal and hepatic? If yes, it might be worth including in the manuscript.

·         “limitation of this study if the inclusion of patients” should be “limitation of this study is the inclusion of patients”

Author Response

The study by Raynes et al aims to examine how the inflammatory status may be related to irAE and/or survival in patients with NSCLC expressing PD-L1 ≥50% who received first-line pembrolizumab monotherapy. Authors report several interesting finding including associations between prolonged PFS and OS and dermatological, endocrine, gastrointestinal and hepatic, but not pulmonary, irAEs. Mild irAEs were associated 37 with better PFS and OS in all patients, including on time-dependent analyses. Also, the occurrence of irAEs predicted favorable outcomes regardless of baseline inflammatory status. Finally, authors report that the individual prognostic value of irAE and biomarkers of systemic inflammation is a confounder in associations between these two factors. The manuscript is well powered and well written. This reviewer has no major concern with publication of this manuscript.

The authors thank reviewer 2 for their positive feedback. These comments have been taken into account in the updated manuscript.

Minor points/concern:

  • Did authors find any correlation / significant overlap between types of irAEs for example, gastrointestinal and hepatic? If yes, it might be worth including in the manuscript.

Point taken. This has been updated in the manuscript.

Results – Amongst patients with any irAE, those who received steroids had poorer OS than those who did not receive steroids (HR 1.95 (95%CI 1.04-3.67), p=0.037), but no significant differences between PFS was observed (Supplementary Figure 3). When only patients with mild irAEs were considered, steroid use was not associated with differences in PFS or OS. When treatment discontinuation due to irAEs was considered as a variable, patients with any irAE who discontinued treatment had poor PFS (HR 1.76 (95%CI 1.03-3.01), p=0.041) and OS (HR 2.00 (95%CI 1.10-3.65), p=0.024), but again these differences were not observed with respect to mild irAEs (Supplementary Figure 4). Amongst 17 patients who experienced multiple irAE 12 (71%) experienced at least one endocrine irAE. No statistically significant differences in survival were observed between patients with single or multiple irAE.

  • “limitation of this study if the inclusion of patients” should be “limitation of this study is the inclusion of patients”

Point taken. This has been amended in the manuscript.